# A VPN Performances Analysis of Constrained Hardware Open Source Infrastructure Deploy in IoT Environment

Antonio Francesco Gentile [1], Davide Macrì [1], Floriano De Rango [2], Mauro Tropea [2,*] and Emilio Greco [1]

1   Institute for High-Performance Computing and Networking (ICAR), National Research Council of Italy (CNR), Via P. Bucci, 8/9C, 87036 Rende, Italy
2   DIMES Department, University of Calabria, Via P. Bucci 39/C, 87036 Rende, Italy
*   Correspondence: m.tropea@dimes.unical.it; Tel.: +39-0984-494786

**Abstract:** Virtual private network (VPN) represents an HW/SW infrastructure that implements private and confidential communication channels that usually travel through the Internet. VPN is currently one of the most reliable technologies to achieve this goal, also because being a consolidated technology, it is possible to apply appropriate patches to remedy any security holes. In this paper we analyze the performances of open source firmware OpenWrt 21.x compared with a server-side operating system (Debian 11 x64) and Mikrotik 7.x, also virtualized, and different types of clients (Windows 10/11, iOS 15, Android 11, OpenWrt 21.x, Debian 11 x64 and Mikrotik 7.x), observing the performance of the network according to the current implementation of the various protocols and algorithms of VPN tunnel examined on what are the most recent HW and SW for deployment in outdoor locations with poor network connectivity. Specifically, operating systems provide different performance metric values for various combinations of configuration variables. The first pursued goal is to find the algorithms to guarantee a data transmission/encryption ratio as efficiently as possible. The second goal is to research the algorithms capable of guaranteeing the widest spectrum of compatibility with the current infrastructures that support VPN technology, to obtain a connection system secure for geographically scattered IoT networks spread over difficult-to-manage areas such as suburban or rural environments. The third goal is to be able to use open firmware on constrained routers that provide compatibility with different VPN protocols.

**Keywords:** VPN; IPsec; SSTP; OpenVPN; OpenConnect; Accel-PPP; Mikrotik; Linux; Windows 10/11; iOS; Android; Mac OS X; OpenWrt; IoT; Libreswan; Strongswan IKE; TLS; SSL





## 1. Introduction

Modern IT infrastructures can cover large geographic areas, and therefore, secure and reliable IT infrastructures are needed, while also guaranteeing low-cost factors, both in terms of space and time. The virtual private network (VPN) is one of the most reliable technology to satisfy this type of need, passing both through the "old" (PSTN) and through the most modern 4G/5G architectures [1–3].

VPNs create a secure connection between the user client and the remote endpoint. Traffic is encrypted to protect it from others. All this takes place via the VPN servers, to which the user's internet traffic is forwarded before reaching its destination [4]. This allows to use public Wi-Fi connections more quietly. Such connections can be easily intercepted, but thanks to a VPN you can avoid nasty intrusions while connected from a hotel, a shopping center or at the airport waiting for your next flight, especially accessing sensitive services, such as your home banking. In this case, it is the encryption offered by the VPN that makes the difference. A tunnel is built between the VPN client and the server on the public network to protect the connection and guaranteeing security and privacy for the users. The connection is initiated by the client that communicates with the server and, once the connection is created, the tunnel is established and the messages are encrypted between the two authenticated parties of the VPN link.

Many types of VPN protocols have been proposed offering different security levels and features. So they present also different performance in terms of latency and throughput output parameters. The choice of the correct VPN solution is fundamental for the performance of the system. Moreover the compatibility with the hardware is important for the user experience. A clear separation between the protocols used on IPsec [5,6] or SSL [7,8] can be defined as follows. In this case, the different typologies of IPsec considered in our work are listed :

- IKEv1-XAUTH, IKEv1-L2TP, IKEv1 RSA
- IKEv2 RSA
- IKEv1 SITE TO SITE PSK, SITE TO SITE RSASIG
- IKEv2 SITE TO SITE RSASIG, IKEv2 SITE TO SITE PSK
- IKEv2 XFRMI ROUTE BASED

    For the considered VPN SSL based [8,9]:

- OpenVPN: ROAD WARRIORS, SITE TO SITE, SITE TO MULTI SITE
- Accel-PPP: SSTP ROAD WARRIORS, SSTP SITE TO SITE, SSTP SITE TO MULTI SITE
- Wireguard: SITE TO SITE, SITE TO MULTI-SITE, ROAD WARRIORS
- Ocserv : SITE TO SITE, ROAD WARRIORS

The main contribution of this work focuses on demonstrating the power of Open-WRT Open Source firmware to implement different types of VPN infrastructures having a native Linux operating system "under the hood" with the possibility of installing it on various boards, from enterprise ones (Motherboard APU) to those TP-LINK/D-LINK for constrained environments and with poor connectivity problems. The main goal is to build an Open Source infrastructure for Smart Devices in IoT Environment for deployment in outdoor locations with poor network connectivity. The OpenWRT firmware allows to manage IPsec VPN initiator/responder (Xauth, L2TP, X509 IKEV1, and IKEV2), VPN client/server for OpenVPN, OpenConnect (aka Cisco AnyConnect) SSTP (Microsoft replaced the old and now insecure PPTP) and Wireguard. It also allows using dynamic routing management systems through software such as Quagga ("Quagga Routing Suite", https://www.quagga.net/ (accessed on 15 June 2022)/Babeld ("Babel Routing Suite", https://www.irif.fr/~jch/software/babel/ (accessed on 15 June 2022))/FRR ("FRRouting Routing Suite", https://frrouting.org/ (accessed on 15 June 2022)). These software packages support protocols such as OSPF ("OSPF Routing Protocol", https://www.cisco.com/c/en/us/td/docs/ios-xml/ios/iproute_bgp/configuration/xe-16/irg-xe-16-book/configuring-a-basic-bgp-network.html (accessed on 15 June 2022)) and BGP ("BGP Routing Protocol", https://www.cisco.com/c/en/us/td/docs/ios-xml/ios/iproute_bgp/configuration/xe-16/irg-xe-16-book/configuring-a-basic-bgp-network.html (accessed on 15 June 2022)), thus enabling the system to create mesh networks realized through point-to-point and point-to-multipoint VPN. Once the infrastructure has been built, the Mosquitto software ("Mosquitto Service", https://mosquitto.org/ (accessed on 15 June 2022)) can be installed to manage the IoT part. Mosquitto is a very light service and can be configured both as a client and as a broker, and therefore has a general-purpose functionality concentrator available in a single device.

This paper has the following structure: Section 2 shows related works about VPN implementations and analyses present in the literature; Section 3 presents the components of a VPN architecture; Section 4 provides the deployed scenarios used for the testbeds; Section 5 provides the experimental results in the implemented scenarios; lastly, Section 6 summarizes the paper.

## 2. Related Works

Many works exist that propose security systems at different protocol stack layers and in different network typologies, showing cryptography performance analysis such as in [10] where authors provide a MAC layer security study on Wireless Sensor Networks (WSNs) also with the use of new technology as Elliptic Curve Cryptography (ECC) [11] proposing

different solution at different network attacks such as denial of services attack [12,13]. Many studies are proposed that deal with the VPN issues showing performance evaluation in terms of throughput and delay in different testbed environments. In this section, we show a brief state-of-the-art summary of the VPN literature in order to present a brief overview about this important topic.

In [7] a general VPN study approach is provided. The work focuses on the concept that VPN technology can only be meaningfully analyzed by referring to the values and motivations of the people who make up companies. A key finding reports the observed differences in "how" the terms "VPN" "security," and "privacy" are perceived and understood in the corporate employee population.

In [14], network layer-based VPN is evaluated. VPN provides secure encrypted communications between remote LANs around the world using Internet Protocol (IP) tunnels and a shared medium such as the Internet. The document focuses in particular on showing the strength of each VPN, through several studies on state-of-the-art and then analyzing the protocol Wireguard. A series of comparisons between deployments based on IPsec and GRE [15] and Wireguard is discussed in [15]. SSL VPN works at secure sockets layer. Again in this work [16], it is highlighted how an SSL-VPN is used extensively to guarantee authentication and confidentiality of data between the client (Web browser) and the server (SSL Server).

In [17] an approach to using VPN technologies for secure encryption of traffic in untrusted networks, such as bars, lounges, conferences, free hotspots, free wifi at the airport, and generically during any trip are discussed.

In [18,19] VPN technology is discussed in depth. In particular, this work focuses on the weaknesses and poor performance in the presence of a high load of a web browser-based SSL VPN, which among other things, only supports Windows [16] operating systems.

In paper [20] the choice of the IPSEC-IKEv2 suite is justified against other (SSL/TLS VPN, SSH Tunnel) possible implementations to ensure the security of the IP communication layer. The motivation lies in the ownership of "transparency" for the layers higher than the IP of IPSEC-IKEv2, as well as allowing a quick update of the encryption in case of data channel compromise. In this work [21] we evaluate the use of IPSEC to create VPNs suitable for "e-business" needs. A router Linksys, with the functionality of both an access point and a VPN concentrator and OpenSWAN as a system daemon, is used. This router supports the IKEv1-L2TP tunnel type (authentication/traffic via L2TP and encryption guaranteed by IPSEC).

The IPSec stack consists of three subcomponents: Encapsulating Security Payload (ESP), Authentication Header (AH), and Internet Key Exchange Protocol (IKE). The AH protocol is also discussed in [22] the security and quality of service aspects of VPNs are considered. The IKE protocol allows the various VPN endpoints to produce session keys for secure communications through a series of message exchanges. IKE currently consists of two versions, IKEv1 [23,24] and IKEv2 [16]. IKEv1 is very flexible and allows different configuration options, but it has a great architectural complexity that makes it increasingly difficult to deploy in modern networks. IKEv2 collects the legacy of the previous version and overcomes its limits, also providing more modern and performing encryption systems for network traffic. An important feature of IKEv2 is native support on current platforms (OS X 10.11+, iOS 9.1+, Linux 3.x + and Windows 8+). On the Mobile front, IKEv2 is supported both by native clients and with third-party apps on iOS, Android, Blackberry, and Windows devices.

QUIC is a UDP-based transport protocol, particularly suitable for IoT devices with limited resources [25,26] and general TRANSPORT SERVICES (TAPS) [18] implementation guidance can be found in [20]. Ref. [25] discusses Layer 3 VPN Tunnel Traffic Leakages and discusses possible mitigations.

In the work [19] we deal with the issue of the parameters to be provided to authenticate to a VPN server. Mainly, in many cases, the classic coupled username/password valid is

enough to access, while for other implementations, it is necessary to provide additional parameters, as in the case of an IKEv1-XAUTH deployment.

The Secure Socket Tunneling Protocol (SSTP) [27,28] proposed by Microsoft allows channeling the traffic of a PPP device through an encrypted VPN tunnel via HTTP using SSL/TLS transport. This component, in particular, deals with negotiating keys, encrypting the channel, and ensuring the integrity of the traffic.

OpenConnect, discussed in [26], is an software designed to create point-to-point and point-to-multi-point secure architectures. Born as an open-source implementation of Cisco's AnyConnect SSL VPN client, it guarantees a good level of protection and data exchange at the application level [29]. This result allows to securely manage communications between applications and services running on IoT nodes and on cloud/edge infrastructures.

With the term "VPNaaS," we mean a simple way to parameterize network security, centralizing the necessary configurations in a single product to meet numerous corporate security requirements and access to virtualized networks on the cloud. In particular, this work [30] proposes the implementation of such a service using WireGuard with a WAN backbone based on 5G transport.

In [22], experimental analysis was performed on Debian environment Linux by implementing the IPsec tunneling protocol with different encryption algorithms. The paper concludes that IPSec AES-sha1 provides fair and reasonable terms performance comparable to IPSec 3DES-sha1. It is also underlined how the encryption/decryption of the UDP VPN (User Datagram Protocol traffic) requires a large amount of CPU and memory that contributes to performance degradation.

The document [31] compares the VPN technologies "L2TP", "PPTP", "OpenVPN", "Ethernet over IP" (EoIP) and "MPLS". This analysis stemmed from having to choose the technology that best-suited business needs. For each VPN technology examined, accurate analysis of both performance and packets in transit was made.

## 3. Components of VPN Architecture

This section introduces the two currently most popular VPN protocols / implementations (IPsec, VPN based on SSL/TLS). They represent both the state-of-the-art and a starting point for new research in the sector and are supported both in commercial (e.g., Mikrotik) and open source products (e.g., OpenWrt, Linux Distros). In the following we present the used protocols for VPN implementation, the Linux OS for implementing the VPN and the considered hardware platforms, proprietary and free.

### 3.1. IPsec

IPsec is used to manage encrypted VPN tunnels at OSI Layer 3 [5,6]. IPsec is part of a series of protocols and its architecture has been proposed as a standard by the IETF, an organization that is responsible for continuing the technical development of the Internet. It was created for the new version of the network protocol (IPv6) and later also for IPv4. It can be configured according to three functional implementations:

- Transfer protocols : Encapsulating Security Payload (ESP) and Authentication Header (AH)
- Encryption Process Management: Internet Key Exchange (IKE) and Internet Security Association and Key Management Protocol (ISAKMP)
- Database: Security Policy Database (SPD) and Security Association Database (SAD)

Using the AH and ESP protocols, IPsec makes it possible to guarantee the integrity and authenticity of the data sent. The AH protocol, via the Packet Accelerator Extension, authenticates the data source and protects against modification of packets during transmission. Finally, it adds to the header a sequence number for preventing packets from being sent.

The ESP protocol guarantees, in addition to verifying the identity and integrity of the data, also the encryption of the pending data. It should be noted that ESP authentication does not consider the most external IP header and therefore it is not complete, therefore

an additional encapsulation is required because the ESP contents are delivered correctly, traversing networks connected via Network Address Translation (NAT), as is generally the case in private xDSL networks. IPsec is configured to work in two modes: Tunnel and Transport.

By using "transport mode", the corresponding transfer protocol is added between the IP packet header, which remains unchanged, and the data area. The protection starts on the outgoing computer and comes up until it reaches the target computer. After the package is received, the original data are decompressed and made available. Transport mode has a very low elaboration time, and guarantees data security only to the detriment of that source and destination addresses. This is used for "Host 2 Host" or "Host 2 Router" connections.

By using "tunnel mode", the data packets receive a fresh complete IP header, then original address, destination address, and data are hidden. In addition to this the header of the respective transfer protocol is generated, creating the so-called "encapsulated" mode. The new outermost IP header defines the endpoints, encrypted and differentiated from the communication endpoints fixed in the actual IP header. Only when the packet has been unpacked on the encrypted endpoints it will be forwarded to the recipients. Data transmission in tunnel mode occurs at "Site 2 Site", "Host 2 Site", and "Host 2 Host".

### 3.2. SSL/TLS Based VPN

SSL/TLS VPN (Secure Sockets Layer VPN) provides a standard VPN solution based on a Web browser in Transport Layer or developed using specific client/server applications [7,8]. Sockets are used for data transfer between sender and receiver. There are two general deployments for SSL/TLS VPN implementation.

SSL/TLS Portal VPN - SSL/TLS Portal VPN: In this scenario, secure service access is achieved with a single standard SSL/TLS connection to the target web server. The client can access the SSL/TLS VPN gateway via a standard web browser, providing the necessary parameters to negotiate authentication [9].

SSL/TLS Tunnel VPN: In this scenario the VPN client can access multiple network services/hosts (this is the case with OpenVPN, OpenConnect, and SSTP). In the classic SSL/TLS communication, two keys are used to encrypt the data, the public one, shared among all, and the private one, typical of each endpoint. To further increase the level of security, two-factor authentication can be used by far or also OTP. In IPsec communication, once the client is authenticated to the VPN, it has full access to the private network, while in SSL/TLS VPN you have control of the more granular access, which allows you to create specific tunnels for applications via sockets rather than for the entire network, also creating specific "access roles" (access profile with specific rights for different users).

### 3.3. Linux Daemons

A VPN based on IPsec on Linux systems consists of the implementation via SW of the IKE, AH, and ESP protocols, using appropriate modules made available by the kernel.

IKE: As the name indicates, the purpose of the IKE protocol is to authenticate (using a pre-shared key, typing a public key, freeradius) the VPN peers, dynamically generate the keys, and share them with VPN peers. Keys are also used for the second phase of IPsec from IKE. Libreswan implements the IKE protocol using the project's bar program. ESP: The ESP protocol is the actual specification of the policy agreed by colleagues which is implemented in the Linux kernel's IPsec stack (NETEY/XFRM).

#### 3.3.1. Libreswan/Strongswan

LibreSwan [32] and StrongSwan [33] are open source implementations of the IPsec protocol, powered by OpenSwan and based on the FreeSwan project, available as a ready-to-use package on Linux distributions on RedHat. However, detailed instructions are provided in the project's code for compiling on non-packaged Linux platforms. After the installation process, after the proper configuration, one will have an IPsec VPN gateway

capable of protecting data in transit between the members of the network. Below, Table 1 compares LibreSwan and Strongswan features.

**Table 1.** Strongswan vs. Libreswan comparison Table.

| Feature | LibreSwan | Strongswan |
| --- | --- | --- |
| Pre-shared key authentication | Yes | Yes |
| Public-key authentication | Yes | Yes |
| IKEv1 key exchange | Yes | Yes |
| IKEv2 key exchange | Yes | Yes |
| AH support | Yes | Yes |
| NSS cryptographic library | Yes | No |
| Xauth and DNSSec | Yes | Yes |
| Network Manager | Yes | Yes |
| Virtual IP Addresses | Yes | Yes |
| MOBIKE | Yes | Yes |
| NAT Traversal | Yes | Yes |
| Route-based VPN | Yes | Yes |
| Policy-based VPN | Yes | Yes |
| Policy-based VPN and Route-based VPN simultaneous | Yes | No |
| Enable weak ciphersuites for backwards compatibility | No | Yes |
| High Availability | Yes | Yes |

Libreswan configuration files are not compatible with strongSwan ones, although the format of ipsec.conf is identical. Different options have different meanings, others are mutually absent due to the supported architectures. For example, in Libreswan it is not possible to enable the L2TP IPsec [25] support of Android, because it is fixed in the client with the DH2 version (MODP1024), considered too weak to be supported, and it is disabled in the compilation phase. Strongswan and Libreswan both support both Policy-Based and Route-Based VPN, but in the former case they can be used in a mutually exclusive way unless you switch from the "classic" configuration (ipsec.conf) to the new one (swanctl.conf), while in the second, there are special flags, both policies are supported by default.

### 3.3.2. Accel-PPP

Accel-PPP [34] is a VPN concentrator designed to have high performance for Linux systems and allows the user to manage standard VPN technologies with a single application. Many open source projects provide VPN services but specialize in a specific technology. With Accel-PPTP, one has an all-in-one system with centralized configuration, management, and monitoring. Accel-PPP allows managing protocols: PPTP PPPoE L2TPv2 SSTP IPoE. The accounting is configurable via file or the use of Radius services. Authentication is managed through the mechanisms: PAP, CHAP (md5), MSCHAP-v1, and MSCHAP-v2 extensions, while EAP is not supported. All PPPoE, PPTP, and L2TP tunnels use special kernel modules to optimize performances.

### 3.4. Operative Systems and Hardware Platforms

In the current panorama, many operating systems and HW/SW platforms are dedicated to networking and secure communications. We have: MikroTik develops MikroTik RouterOS, the operating system of RouterBOARD boards; OpenWrt, developed by OpenWrt Project, an embedded Linux operating system, used on embedded devices for routing network traffic; PfSense, based on FreeBSD, an open source router and firewall with features that allow to manage unified threat, multi WAN and load balancing; OPNsense, another FreeBSD-based open source firewall that guarantees high security, Intrusion Prevention, Traffic Shaping and Captive Portal services; IPFire, based on Linux, a distribution designed

to use the machine as an internal or perimeter firewall; VyOS, based on open Linux and distributed by the Sentrium company. It is open source and designed to protect the network and corporate data with high performance; Gargoyle, a firmware designed to use the machine as an internal or perimeter firewall; LibreMesh, an Open Source Sofware for Geek-free Mesh Community Networks. In this article, we will focus on the OpenWRT firmware and the Mikrotik platform for comparison.

### 3.4.1. OpenWRT

OpenWRT [35] is a distribution originally intended for use on wireless routers, to extend their functionality over manufacturer-supplied firmware. The operating system guarantees a filesystem with write permissions by the user, allowing among other things the installation of third-party software and therefore the possibility of extending its functionality.

This allows you to make use of the most recent routing software, and guarantees greater security and fewer bugs than pre-installed stock manufacturer software, especially in older devices no longer supported.

It can also be installed on custom HW, such as specific boards, and in the case of x86-64 platforms, it also supports virtualization, allowing small networks of containers for ad-hoc services, as well as providing network access to all devices in the LAN.

### 3.4.2. MikroTik

MikroTik [36] is a Latvian company based in Riga and produces equipment for networking and internet connectivity, in particular routers and wireless broadband equipment for Wireless ISPs. It is present in almost all countries of the world.

MikroTik's experience in creating hardware and routing systems highly compatible with the most widespread industry-standard systems led to the creation in 1997 of the RouterOS software, with high control, and flexibility for all types of routers and interfaces, developed on the Linux kernel. Thanks to RouterOS any PC or MikroTik RouterBOARD can become a dedicated router. Being proprietary devices, the flexibility in installing packages for additional features, however respectable (supports IoT, Lora, and 4G/5G), is lower than that of OpenWRT. Figure 1: highlights the ability of a Mikrotik router to act as a "Publisher" to an MQTT Broker (deployed on an OpenWRT router).

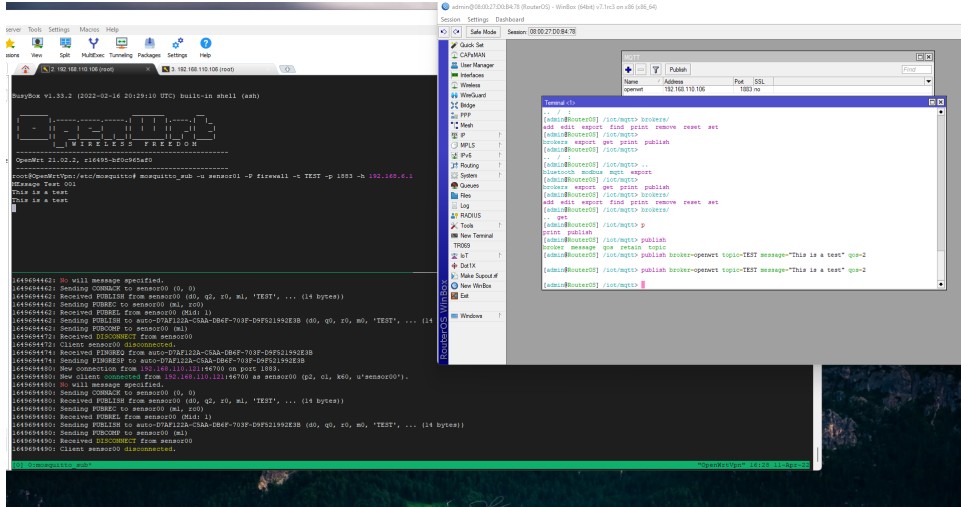

**Figure 1.** A publish/subscribe scenario from Mikrotik to OpenWRT MQTT Broker.

## 4. Description of Different Deployed VPN Solutions

To create a VPN, there are different solutions, both HW and SW (e.g., use Mikrotik RouterBoards, ("Mikrotik ", https://mikrotik.com/ (accessed on 15 June 2022)) or implement the appropriate stack on your own server, in this case with Debian OS ("DEBIAN OS " https://www.debian.org/index.it.html) (accessed on 15 June 2022)), with peculiar properties depending on the chosen implementation. In particular, we will focus on three macro application scenarios:

- A Site to Site scenario: (where two remote offices communicate securely), shown in Figure 2
- A Site to Multi-Site scenario: (where multiple remote offices communicate securely), shown in Figure 3
- A Road Warriors scenario: (in which multiple users from remote offices communicate securely with a particular local office), showed in Figure 4.

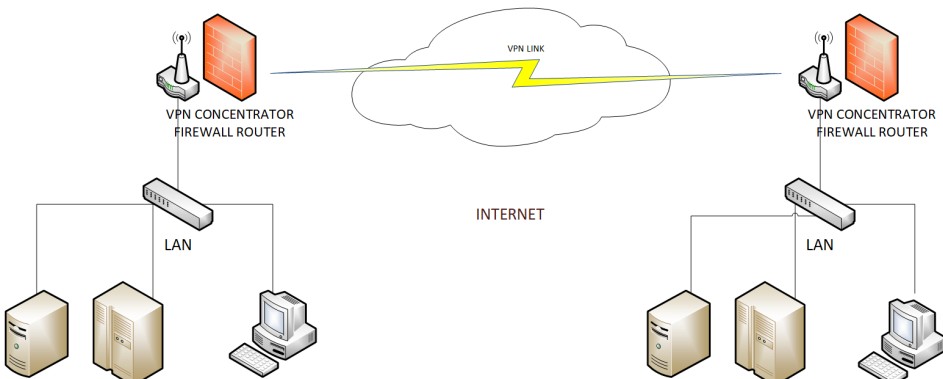

**Figure 2.** Site to Site VPN Scenario.

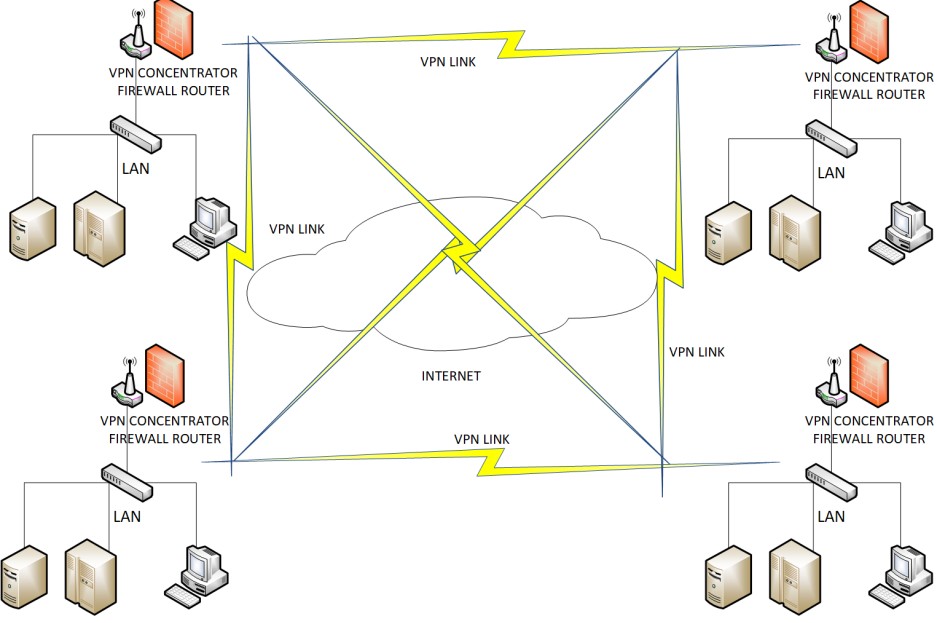

**Figure 3.** Site to Multi Site VPN Scenario.

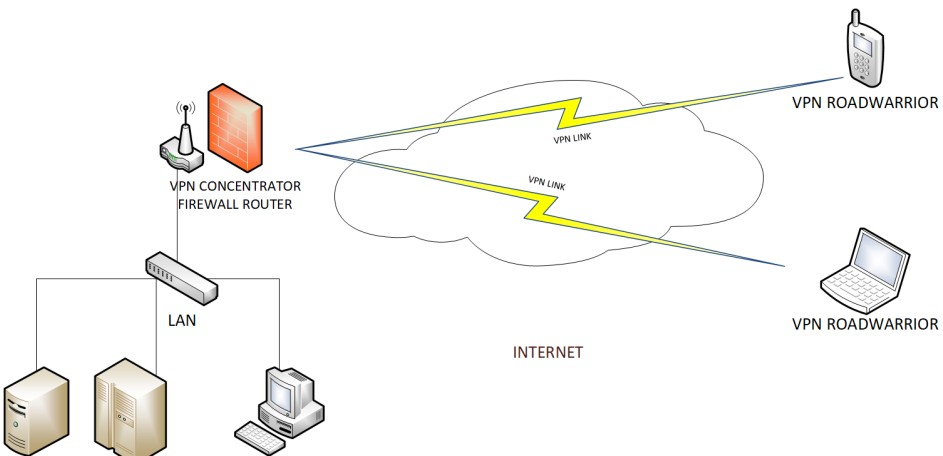

**Figure 4.** Site to Road Warriors VPN Scenario.

These solutions provide encryption and integrity of data in transit by relying on encryption algorithms ranging from Blowfish to AES (128–512) up to implementations that use elliptic curves (e.g., ecp192). Depending on the type of device/network to be interconnected, various alternatives are available, especially in the case of Road Warriors (RW), passing from authentication via PSK + XAUTH (e.g., in IPsec) to that through certificates, or even to two-step solutions, through one-time password (OTP) or supplementary credential files (case RADIUS + VPN IPsec/OpenVPN/OpenConnect). In this work we wanted to make a "field" assessment of the current landscape both at the level of communication infrastructures (WIFI, CABLE 4G/5G), and at the level of devices that it is possible to connect (NUC, mini PC, Raspberry Pi, Smartphones). Involved variables are many, ranging from guaranteeing the widest support to the greatest number of devices and technologies, trying to guarantee the best performance with the same safety.

Table 2 provides a summary of the implemented testbeds: from a description of the network infrastructure to the software used.

Many of the architectures proposed were built in the projects mentioned above and are still in production. OpenVPN and Wireguard are part of the Cogito project to interconnect the cloud and local offices. One used IPSec in the Domus project to manage data exchanges and resource access policies. Finally, one used OpenConnect and OpenVPN in the RES-NOVAE project for the same purpose.

Moreover, given the exponential increase of technologies such as the IoT [37], and the consequent development of communication paradigms such as "Fog Computing" or "Cloud oriented approaches" (e.g., AMAZON AWS), or multisite clusters (e.g., realized through Kubernetes [38]) both on fiber/PSTN carriers and on 4G/5G carriers, this ever-growing volume of data, also in compliance with the recent "General data protection regulation" (GDPR) regulations, must be managed properly. Another great advantage, deriving from the use of the OpenWRT ("OpenWRT Project ", https://openwrt.org/ (accessed on 15 June 2022)) operating system, allows the implementation of these technologies on all models of compatible routers and devices. The wide range of supported "bridges" allows, especially in the case of IoT networks, to interface with third-party protocols, to be used as a perimeter gateway, such as LoRa and Zigbee, as well as the classic ones supported by TCP/IP stack. If you consider that the x86-64 versions also easily support virtualization, you can create Fog networks of micro-services and IoT multi-protocol with negligible figures and connect them securely. An example could be given by the installation of outdoor sensor networks on areas covered in WAN by 4G/5G networks and locally configurable and processable with an "Edge Computing" approach.

**Table 2.** A summary of the implemented testbeds.

| VPN Software Managed | VPN Deploy | VPN Implementers | Operative Systems | Platforms |
|---|---|---|---|---|
| IPsec LIBRESWAN 3.27 STRONGSWAN 5.9.1 | ROAD WARRIORS | IKEv1-XAUTH IKEv1-L2TP IKEv2 RSA | LINUX DEBIAN 11 WINDOWS 10 (client) WINDOWS 11 (client) DEBIAN 11/10 (client) ANDROID 11 (client) iOS 15 (client) MAC OS X 13 (client) RASPBERRY Pi 2/3/4 MIKROTIK 7.X OpenWRT 21.x | armv7 ×86 ×86-64 ARM64 ARM MIPSBE MMIPS SMIPS PPC |
| IPsec LIBRESWAN 3.27 STRONGSWAN 5.9.1 | SITE TO SITE | IKEv1 SITE TO SITE PSK IKEv2 SITE TO SITE RSASIG | Same as above | Same as above |
| IPsec LIBRESWAN 3.27 STRONGSWAN 5.9.1 | SITE TO MULTI SITE | IKEv2 XFRMI ROUTE BASED | Same as above | Same as above |
| OpenVPN 2.5.1 | ROAD WARRIORS SITE TO SITE SITE TO MULTI SITE | | Same as above | armv7 ×86 ×86-64 |
| Accel-PPP Latest github stable realease | SITE TO SITE SITE TO MULTI SITE | SSTP SITE TO SITE SSTP SITE TO MULTI SITE | Same as above | Same as above |
| Accel-PPP Latest github stable realease | ROAD WARRIORS | SSTP ROADWARRIORS | Same as above | Same as above |
| Wireguard 1.0.2 | SITE TO SITE SITE TO MULTI SITE ROAD WARRIORS | | Same as above | armv7 ×86 ×86-64 ARM64 ARM MIPSBE MMIPS SMIPS PPC |
| Ocserv 1.1.2 | SITE TO MULTI SITE SITE TO SITE ROAD WARRIORS | | Same as above | armv7 ×86 ×86-64 |

## 5. Experimental Results in Implemented Scenarios

In order to evaluate the performance of different VPN protocols on different operating systems and hardware as represented in Table 2, different topologies were configured as in Figures 2–4. Both Iperf software and Atop are used to record the activities of the server's operating system during the various VPN sessions. Network topologies nodes are shown in Table 3.

**Table 3.** Network topologies HW components.

| Hardware | Quantity |
|---|---|
| Workstations with Intel® i7 @ 1.80 GHz processor, 8GB RAM | 2 |
| VMWare Mikrotik RouterOS x86-64 virtualized | 1 |
| TP-Link Archer C7 v5 Qualcomm Atheros QCA956X v1 128 MB RAM | 1 |
| TP-LINK TL-WR841N (RO) v11 Qualcomm Atheros QCA9533 32 MB RAM | 1 |
| TP-Link TL-SG105E Switch 10/100/1000 Mbps | 1 |

Table 4 shows a comparison of Authentication and Encryption algorithms for deployed scenarios and proposed by default client-side.

**Table 4.** Comparison table of Authentication and Encryption used in the experiments and proposed by default client-side.

| VPN Type | Encryption | Authentication |
|---|---|---|
| *SSTP* | AES-256 | Username/Password, Certificates |
| *IKEv1-L2TP* | AES-256-GCM128 | Username/Password/PSK |
| *OPENCONNECT* | AES-256 | Username/Password, Certificates, OTP |
| *OPENVPN* | AES-256, CHACHA20 | Username/Password, Certificates, OTP |
| *IKEv2 X509* | AES-256 | Certificates |
| *IKEv2 PEAP* | AES-256 | Username/Password, Certificates |
| *IKEv1-XAUTH* | AES-128 | Username/Password, PSK |
| *WIREGUARD* | CHACHA20, CURVE25519 | Certificates/Keys |

The nodes are connected to a 10/100 Ethernet switch with 100 Mbps UTP links. The complete topology consists of a subnet representing the WAN (with public IP assigned, for testing from 4G/5G networks), two private subnets for each workstation, one subnet for Mikrotik and one subnet for OpenWRT, as shown in Figure 5. In the "Site to Site" topology, two VPN servers act as software routers and endpoints of the VPN tunnel and they are connected to local workstations where traffic is generated. In the "Site to Multisite Site" topology, the servers become three (the number of nodes can be increased), they maintain the function of the software router and endpoint of the VPN tunnels and are connected to local workstations on which traffic is generated. In the "Road Warriors" topology, a VPN server acts as a software router and endpoint of the VPN tunnel and it is connected to local workstations on which traffic is generated, this time directed towards mixed clients (smartphones or laptops connected via wireless). To generate the network traffic, the "Iperf" tool was used, in versions 2 and 3 (both UDP and TCP tests). This tool measured productivity and round-trip time (RTT) while several counters of the same operating system measured the CPU usage.

*5.1. Considered VPN Deployed Topology*

Some deploys described in this paper were also used in the context of the "DOMUS" ("Progetto DOMUS ", https://www.gruppotim.it/it/archivio-stampa/mercato/2016/TIM-Distretto-Domus-Cosenza-14Dicembre2016.html (accessed on 10 June 2022)), "COGITO" ("Progetto COGITO ", https://www.icar.cnr.it/progetti/cogito-sistema-dinamico-e-cognitivo-per-consentire-agli-edifici-di-apprendere-ed-adattarsi/ (accessed on 10 June 2022)) and "RES-NOVAE" ("Progetto RES-NOVAE ", https://www.cueim.org/progetti/res-novae-reti-edifici-strade-nuovi-obiettivi-virtuosi-per-lambiente-e-lenergia-smart-city/ (accessed on 10 June 2022)) projects and can be replied in any environment. The reason for the choice of the HW considered is given by the need to measure the performance of components already put into operation in the three research projects referred to above and also to ensure maximum backward compatibility.

The basic network topology with example IP address ranges is depicted in Figure 5. From time to time, the right endpoint is replaced in the site-to-site scenario, and the connections whose data are shown in the figures are established. For Wi-Fi-only devices, acting as a VPN client, the connection is established with the left endpoint. The Wi-Fi network these devices are originally connected to is the dummy WAN.

The dummy WAN is a simple network with a router (IP 192.168.110.254) on the segment 192.168.110.0/24, which simulates a public network and allows communication between VPNs endpoints (connected wired / wireless clients to the router or switch). In particular, two private IP of the fictitious WAN simulate the public IP of the gateways of two LANs located at a lower level and used to test site-to-site topology, respectively, with "internal" networks 192.168.111.0/24 (site one) and 192.168.112.0/24 (site two). The Wi-Fi clients are connected directly to the access point of the primary router. All the RTT tests were carried out by reaching the local network IP of the OpenWRT router, which acts as a VPN centralizer from the client on duty, as shown in Figure 5.

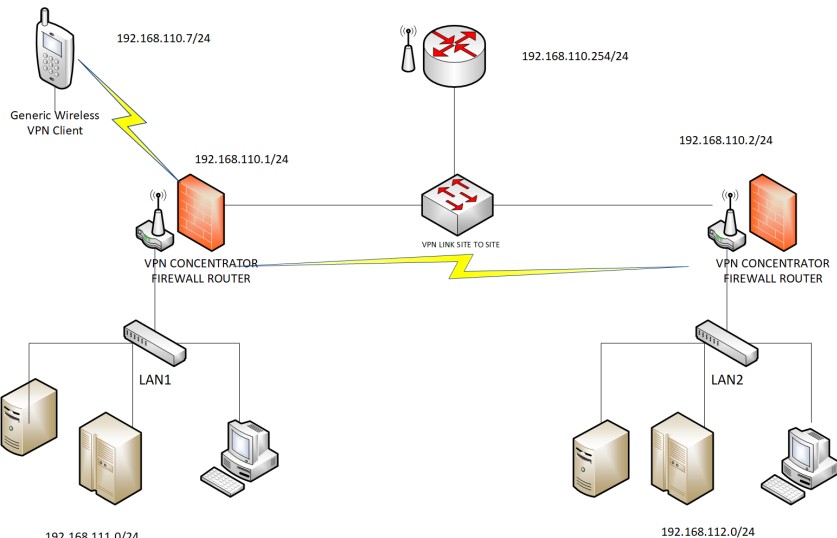

**Figure 5.** Generic VPN Deploy Topology.

Both tools, Iperf2 and Iperf3, were used to get throughput results, and the "ping" tool was used to get RTT data. The graphics were generated using Iperf3 because there were not significant differences with Iperf2.

### 5.2. System Tools for Performance Analysis: Iperf and Atop

Iperf allows to measure the bandwidth available for IP networks. It is available in two versions, Iperf2 and Iperf3, released as open source and thanks to the support for all the most popular operating systems (including mobile ones such as Android and iOS) and the ability to adjust numerous parameters in the analysis of network performance for various protocols, is undoubtedly one of the reference tools for network diagnostics.

Atop is conceived to be a system performance monitor, which works through CLI (Command Line Interface) and can record and report the activity of all operating system processes and its HW/SW components. It allows to analyze the server at runtime and automatically in the long term. It shows the resource usage of all processes and devices and provides the ability to monitor threads within processes and highlights critical lines using colors (red). Moreover, it allows to analyze disk I/O and network usage. Thanks to the kernel module netatop it can acquire TCP, UDP data, and network bandwidth. For each established VPN connection, an instance of atop was launched to collect CPU and network statistics data. These data was passed to *atopsar* to generate readable and archival reports via CSV. With special scripts, these CSVs were tested to create successive graphics using regex and Gnuplot. Subsequently, the aggregated graphs were produced using Excel. All the figures relating to the following topologies have a double scale and display respectively the average throughput (left axis) and the standard deviation (right axis), for wired and wireless endpoints (left figure and right figure).

Without using VPN protocols, the average throughput of the network is 94.1 Mbps, in the fast ethernet sections, with an RTT of 1417 ms. In the Gigabit sections, the throughput is 760 Mbps, with an average RTT of 0.730 ms.

### 5.3. IPsec/L2TP Road Warriors Scenario

The IPsec/L2TP scenario is one of the most common deployments today. It is used for its excellent compatibility, but being outdated, it should be avoided whenever possible. IPsec in transport mode and NAT with multiple connected clients often suffer from performance and stability issues. Many L2TP clients are configured using aggressive/PSK mode, which is problematic from a security point of view. The encapsulation of PPP over IPsec can cause MTU problems, and to remedy this, you force the MTU of clients into a range of 1200–1500 KB, depending on the type of WAN connection. For IPsec/L2TP servers behind

NAT, the registry settings on Windows to allow clients to connect have to be changed. The supported clients are many, all Apple iPhone, iPad, Mac OS X, Android, Linux with a command line, and Microsoft Windows. The server has three components to configure: libreswan for IPsec, xl2tpd for L2TP, and pppd for PPP. Figure 6 shows the comparison of the average throughput and standard deviation in IKEv1 L2TP Road Warriors scenario, wired (on the left) and wireless (on the right).

As seen from the figure, OpenWRT as an Ikev1-L2TP client turns out to be the best-performing, closely followed by Windows 10, where, however, it is necessary to work at the registry level to enable Diffie–Hellman group 14, 2048 bit. As for the wireless clients, the iPhones are slightly better performing than Android.

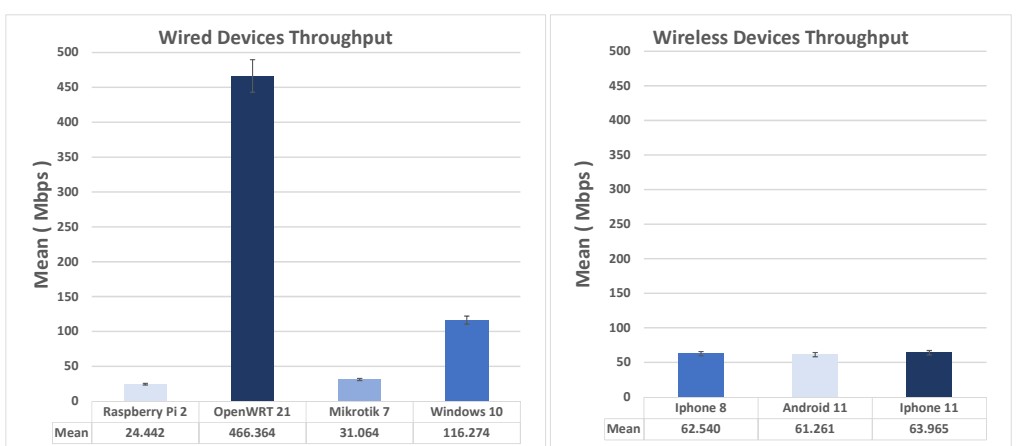

**Figure 6.** Comparison of the average Throughput in IKEv1-L2TP Road Warriors scenario.

*5.4. IPsec Xauth Road Warriors Scenario*

Implementing a VPN service for remote clients using IKEv1/XAUTH with PSK is the least secure way to run IKE/IPsec. The reason is that everyone in the "group" has to know the PSK (also called secret). Even further authentication is required, such as a username and password, someone that knows the PSK can launch a man-in-the-middle attack pretending to be the VPN server. If the client connects to the rogue server, it will tell the attacker their username and password. On Android, this mode is called PSK XAUTH. On iOS/OS X this mode is confusingly called type "IPSec". Some other vendors call it "Group PSK". Supported clients are: all Apple iphones, ipads, Mac OS X, Android, Linux with NetworkManager or commandline and Windows Shrew client, or Cisco routers/firewalls. It is based on draft [19]. XAUTH also requires a username and password. The password can also contain a OTP such as Google Authenticator. Figure 7 shows the comparison of the average throughput and standard deviation in IKEv1 XAUTH Road Warriors scenario, wired (on the left) and wireless (on the right). As seen from the figure, OpenWRT as an Ikev1-XAUTH client turns out to be the best performing, closely followed by Windows 10, where, however, it is necessary to use a third-party application to allow the connection. As for the Wireless clients, the iPhones are the only compatible smartphones.

*5.5. IPsec IKEv2 EAP Road Warriors Scenario*

The following EAP connection definition allows multiple Windows clients to connect to the strongSwan VPN gateway via any EAP method over IKEv2. Cypher suites aes256-sha256-modp2048 for IKE and aes256-sha1-modp2048 for ESP are the strongest proposals; the Windows client is able to offer unless PowerShell is used. These proposals are not explicitly configured here to accept stronger algorithms proposed by other clients via strongSwan's default proposals.

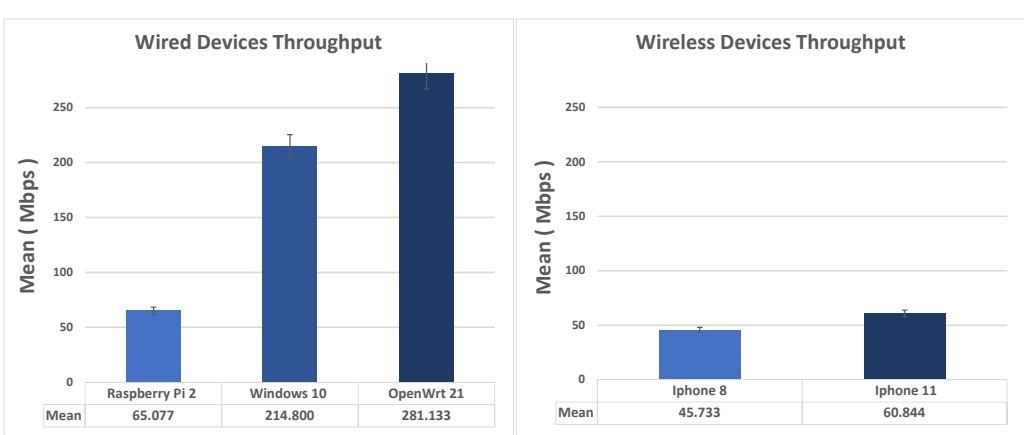

**Figure 7.** Comparison of the average Throughput in IPsec XAUTH Road Warriors scenario.

Figure 8 shows comparison of the average throughput and standard deviation in IKEV2-EAP Road Warriors scenario, wired. As seen from the figure, Windows 10 as an Ikev2-EAP client turns out to be the best performing, while Mikrotik and Raspberry Pi are almost equivalent. The virtual machine running the demo version of Mikrotik is locked to Ethernet 100 Mbps). Using IKEv2 with certificates, the performance of OpenWRT (client) and the Raspberry Pi 2 are noticeably close. As for the wireless clients, the Android, this time, turns out to be slightly better performing than iPhone 11 and iPhone 8.

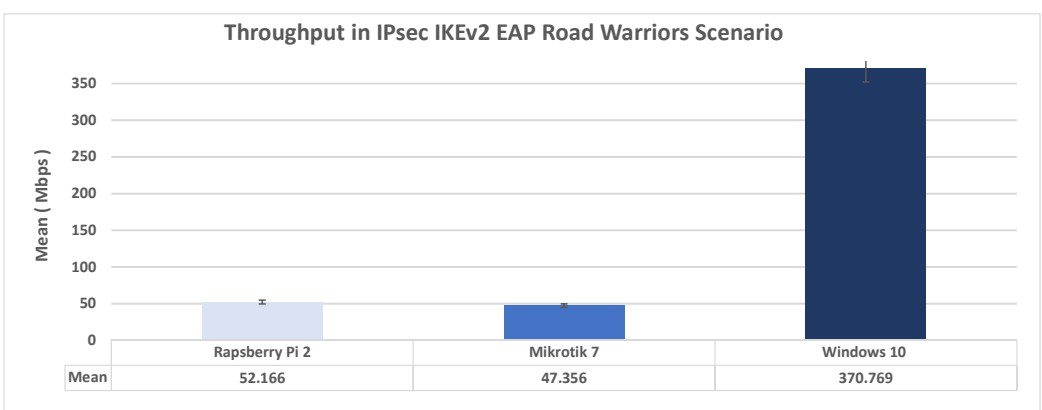

**Figure 8.** Comparison of the average Throughput in IKEv2-EAP Road Warriors scenario.

*5.6. IPsec IKEv2 X509 Road Warriors Scenario*

This method uses IKEv2 without EAP also called "Machine Certificate"-based authentication. Supported clients are:

- libreswan
- Windows 7 and up
- Windows Phone (requires the latest firmware)
- OS X and iOS
- Android with strongswan client.

Figure 9 shows the comparison of the average throughput and standard deviation in IKEv2 Cert Road Warriors scenario, wired (on the left) and wireless (on the right). As seen from the figure, Windows 10 as an Ikev2-EAP client turns out to be the best performing, while Mikrotik and Raspberry Pi are almost equivalent. The virtual machine running the demo version of Mikrotik is locked to Ethernet 100 Mbps). Using IKEv2 with certificates, the performance of OpenWRT (client) and the Raspberry Pi 2 are noticeably close. As for the wireless clients, the Android, this time, turns out to be slightly better performing than iPhone 11 and iPhone 8.

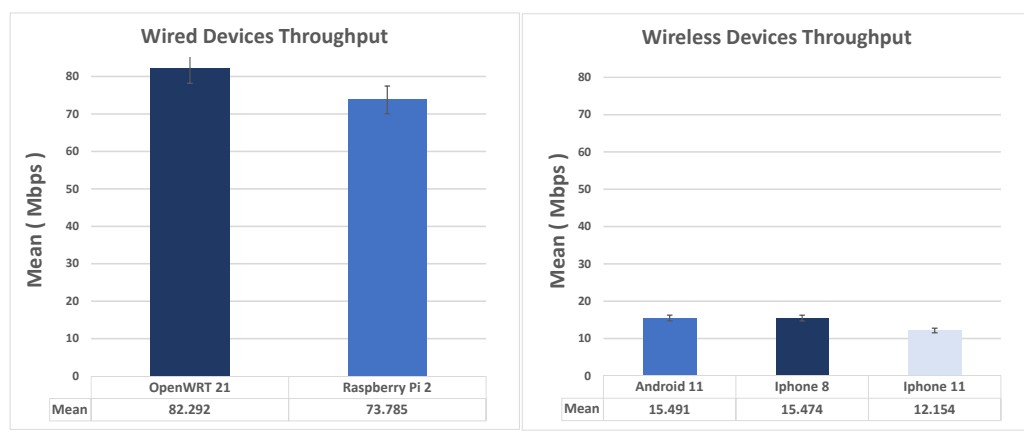

**Figure 9.** Comparison of the average Throughput in IPsec IKEv2 X509 Road Warriors scenario.

### 5.7. OpenConnect (Aka Cisco Anyconnect) Road Warriors Scenario

The OpenConnect protocol is designed to create a channel that makes use of UDP packets and secondarily over TCP if UDP traffic is not allowed. The protocols used by Openconnect are: TLS [RFC8446], Datagram TLS [RFC6347] and HTTP [RFC2616]. The session begins with an HTTP connection over TLS on a known port, after which the client authentication begins. Once this is done, the client initiates an HTTP CONNECT command to establish a VPN channel over TCP and, if possible, a secondary VPN channel over UDP, depending on the response from the server. Once these negotiations are complete, IP packets can travel over the VPN. Figure 10 shows the comparison of the average throughput and standard deviation in OpenConnect Road Warriors scenario, wired (on the left) and wireless (on the right). As seen from the figure, OpenWRT as an OpenConnect client turns out to be the best performing, followed by Windows 10, which uses a third-party application to allow the connection but with a ratio of almost 5:1 and finally from the Raspberry Pi 2, with a nearly 10:1 ratio. As for the wireless clients, the iPhones show inferior performance to Android.

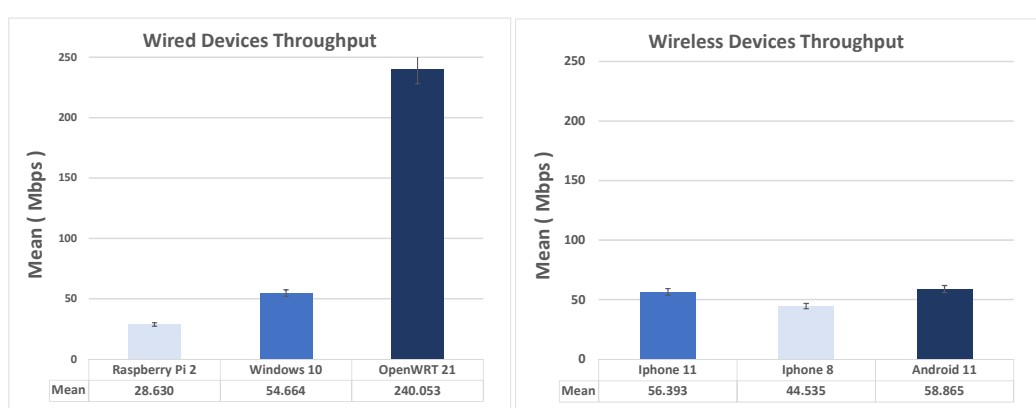

**Figure 10.** Comparison of the average Throughput in OpenConnect Road Warriors scenario.

### 5.8. SSTP Road Warriors Scenario

SSTP, the successor to PPTP-based VPNs, was introduced by Microsoft with the Windows Vista system. This VPN service is still trusted in Windows 7, 8, 10, and 11 versions. SSTP uses 256-bit AES encryption, thus providing reasonable security in data transit while also providing a reasonable speed for encrypted tunnel communications. It is mainly used in the Microsoft environment, but implementations are available on Linux (client/server mode), on OpenWRT (client mode) and on the Mikrotik platform (client/server mode). Figure 11 shows the comparison of the average throughput and standard deviation in Wireguard Road Warriors scenario, wired (on the left) and wireless (on the right). As seen from the figure, OpenWRT as an SSTP client, performs less than

Windows 10, but it still has almost double performance compared to Mikrotik in a virtual machine; both use a dedicated kernel module to establish SSTP connections. Finally, the Raspberry Pi 2 also has noteworthy performance. Windows 10, of course, has the best performance with native support for this technology. As for the wireless clients, the iPhones are incompatible with this technology unless one purchases specific apps.

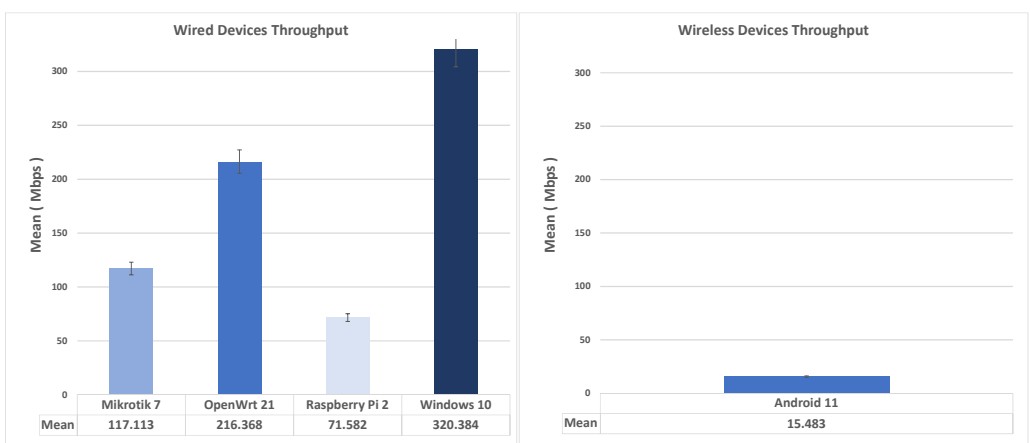

**Figure 11.** Comparison of the average Throughput in SSTP Road Warriors scenario.

## 5.9. OpenVPN Road Warriors Scenario

OpenVPN, a project managed by the "OpenVPN community Team" allows the creation of IP network tunnels using virtual ethernet adapters on TCP/UDP ports. The encryption, authentication, and certification functions are provided by default by the OpenSSL library or by the Gnu-TLS library, and this guarantees a high standard of protection of private network traffic on the public network. OpenVPN allows two authentication modes, one with a static key (PSK) and the other through SSL / TLS certificates that regulate authentication and key exchange using a customized security protocol. Figure 12 shows a comparison of the average throughput and standard deviation in the OpenVPN Road Warriors scenario, wired (on the left) and wireless (on the right). As can be seen from the figure, OpenWRT as an OpenVPN client turns out to be the best performing, followed by Windows 10, which uses a third-party application to allow the connection. Mikrotik only supports TCP-based connections, but guarantees decent throughput. When it comes to wireless clients, Android falls between the two iPhone generations.

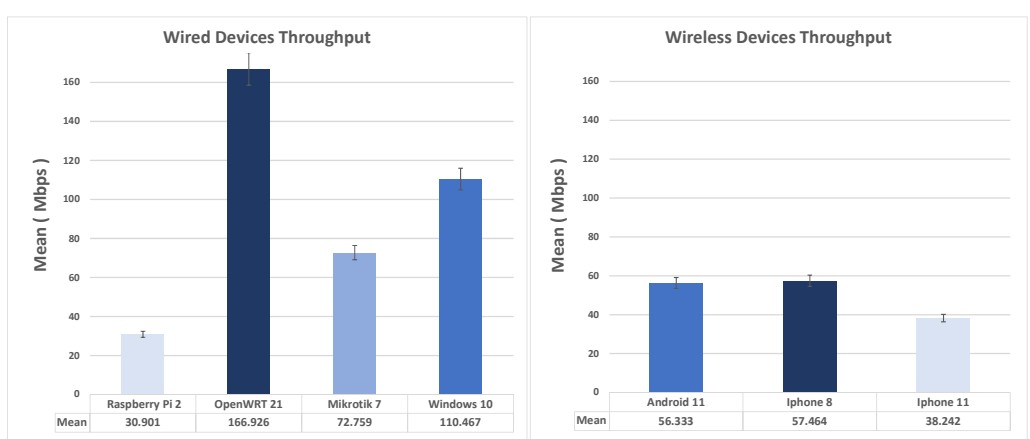

**Figure 12.** Comparison of the average Throughput in OpenVPN Road Warriors scenario.

## 5.10. Wireguard Road Warriors Scenario

WireGuard [39] is a VPN protocol designed to be at the forefront of speed, management, and scalability, with the goal of being leaner than the main contenders, IPSEC and

OpenVPN. It is designed as a general-purpose VPN protocol that can be deployed on both extremely powerful and extremely limited hardware with excellent performance. It is multiplatform and runs both as a client, and as a server on Windows, macOS, BSD, iOS, Android, and Linux (on which it was developed) and is easily distributable. Despite being a young VPN protocol, its qualities, as described in the specific RFC (RFC7539) [40], have already led it to be a reference for the IT communications of the near future. Figure 13 shows a comparison of the average throughput and standard deviation in the Wireguard Road Warriors scenario, wired (on the left) and wireless (on the right). As seen from the figure, OpenWRT as a Wireguard client turns out to be the best performing, followed by Windows 10, which uses a third-party application to allow the virtualized connection. Mikrotik guarantees decent throughput, even if it is worse than the Raspberry Pi 2. Regarding wireless clients, Android falls between the two iPhone generations.

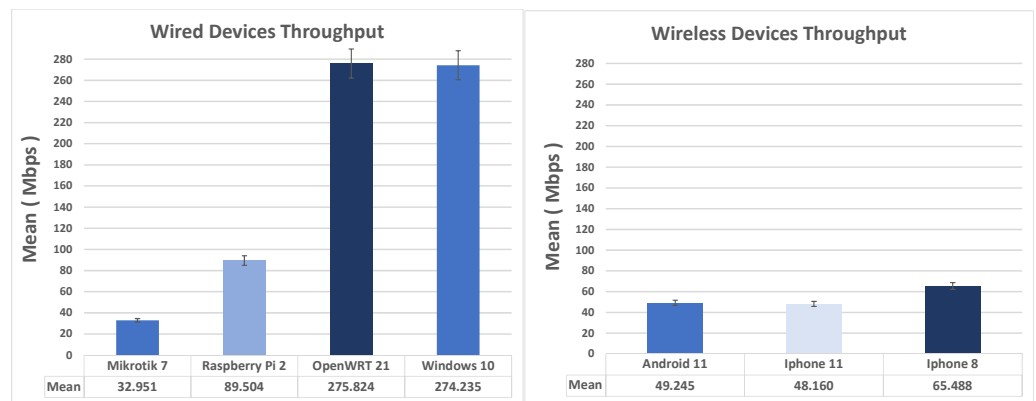

**Figure 13.** Comparison of the average Throughput in Wireguard Road Warriors scenario.

*5.11. Per Device Analysis*

The data analysis performed on each VPN as the devices vary, described in the previous paragraphs, provided helpful information to identify the devices with better throughput, given a pre-existing infrastructure. A complementary analysis of what was described in the preceding paragraphs of this section is addressed in this paragraph. The study, in this case, allows people to highlight, for each device examined, the VPN technology if supported, which guarantees better performance. From Figures 14–20, the following graphs group the average throughput and throughput of the STD data by collecting them by device rather than by VPN protocol type. From a first analysis, it is possible to verify that only a few devices allow the use of all the VPN technologies analyzed, in some cases because the software that implements the technology is missing, in others for reasons of architectural choices of the suppliers. This analysis also gives people a clear view of which VPNs are the best performing and helps the network designer make targeted choices, even in architectures already provided by third parties.

Figures 14 and 15 show the data collected on the average throughput for two different generations of iPhone devices as the VPN established for the iPhone 8 and the iPhone 11 varies. From the graphs, we can see how there are variations in performance between the different VPNs supported. One can observe that for the iPhone 8, the Wireguard and OpenVPN VPNs are the best-performing VPNs, while for the iPhone 11, this is no longer true. Our analysis shows that the IKEv1-L2TP VPN maintains the same performance on average for both iPhone devices tested. Furthermore, from Figure 16, which shows the average VPN throughput per Android 11 smartphone, we can conclude that IKEv1-L2TP VPN is the technology that maintains the highest throughput levels on average across all tested wireless devices. For wired devices, whose average throughput is shown in Figures 17–20. The data collected show a much more heterogeneous situation. For example, from the comparison between Figures 17 and 18, where the average throughput for the Mikrotik 7 device and the Raspberry Pi 2 is shown respectively as the VPN varies, one can

conclude that the choice to connect these two endpoints via a VPN would fall on SSTP technology. While the others perform optimally on one of the two endpoints. Similarly, if we were to connect two endpoints, such as OpenWRT 21, whose average throughput is represented in Figure 19, and the Windows 10 endpoint, whose average throughput is depicted in Figure 20, the choice would have to fall between a VPN between the first three VPN, that have highest throughputs. In our case, the choice would fall on the Wireguard VPN.

Figure 14 shows the comparison of the average throughput and standard deviation in a Road Warriors scenario, using an iPhone 8 client. The best results in terms of throughput are obtained here using Wireguard.

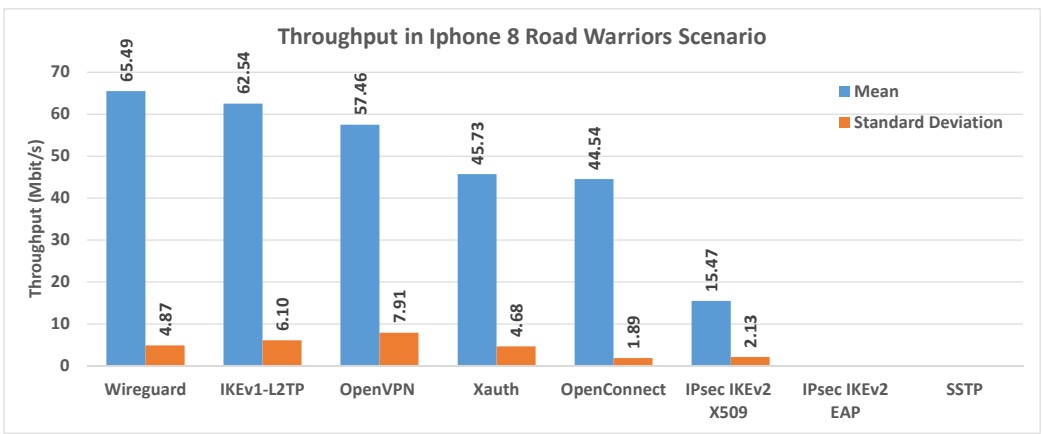

**Figure 14.** Comparison of Throughputs on iPhone 8 in all scenarios.

Figure 15 shows the comparison of the average throughput and standard deviation in a Road Warriors scenario, using an iPhone 11 client. The best results in terms of throughput are obtained here using IKEv1-L2TP.

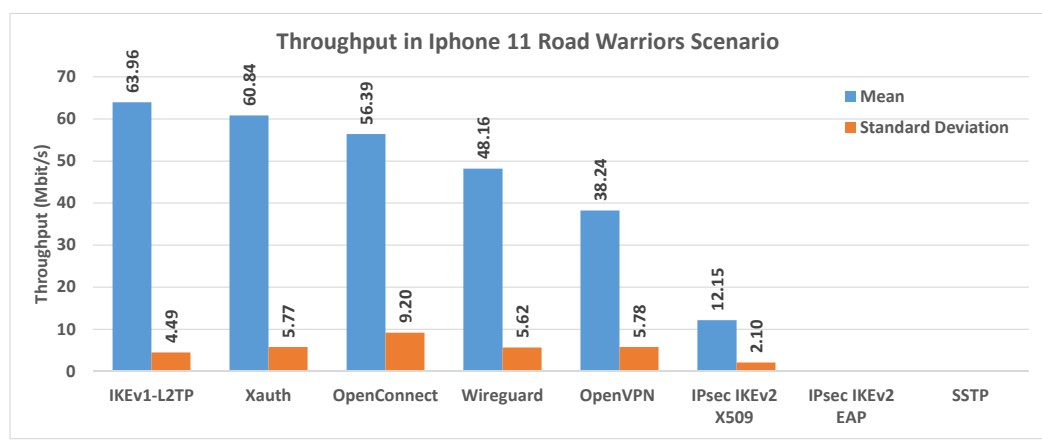

**Figure 15.** Comparison of Throughputs on iPhone 11 in all scenarios.

Figure 16 shows the comparison of the average throughput and standard deviation in a Road Warriors scenario, using an Android 11 client. The best results in terms of throughput are obtained here using IKEv1-L2TP.

Figure 17 shows a comparison of the average throughput and standard deviation in a Road Warriors scenario, using a Mikrotik 7. x endpoint. The best results in terms of throughput are obtained here using SSTP.

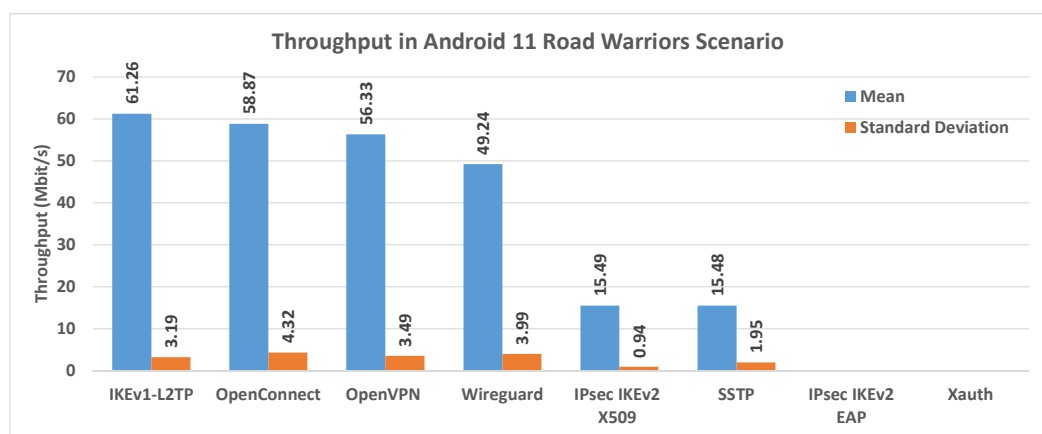

**Figure 16.** Comparison of Throughputs on Android 11 in all scenarios.

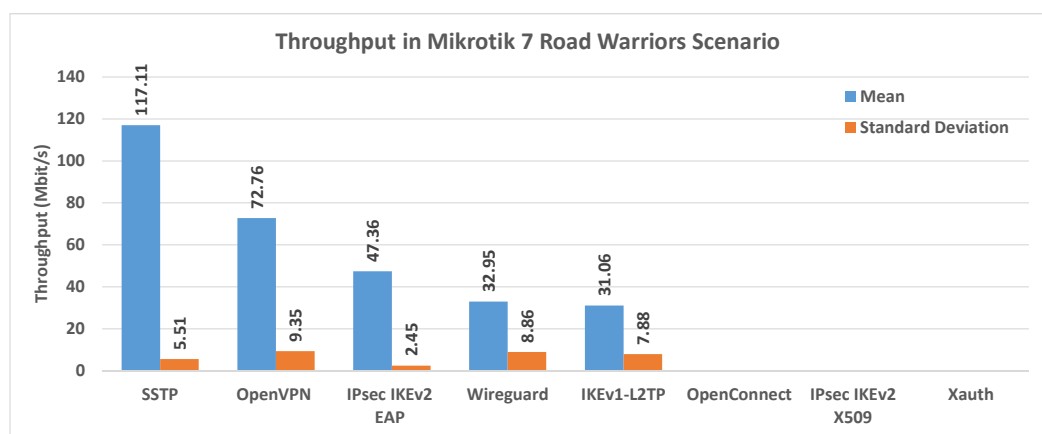

**Figure 17.** Comparison of Throughputs on Mikrotik 7 in all scenarios.

Figure 18 shows the comparison of the average throughput and standard deviation in a Road Warriors scenario, using a Raspberry Pi 2 endpoint. The best results in terms of throughput are obtained here using Wireguard.

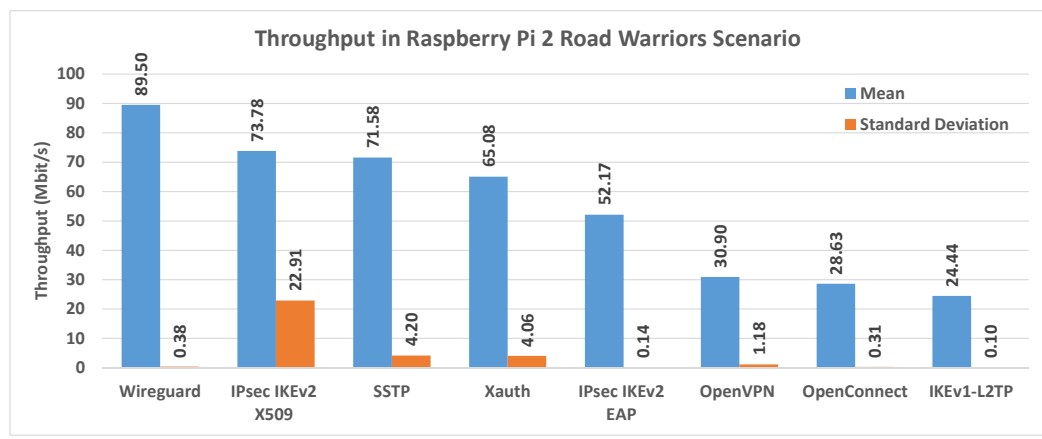

**Figure 18.** Comparison of Throughputs on Raspberry Pi 2 in all scenarios.

Figure 19 shows the comparison of the average throughput and standard deviation in a Road Warriors scenario, using an OpenWRT 21 endpoint. The best results in terms of throughput are obtained here using IKEv1-L2TP.

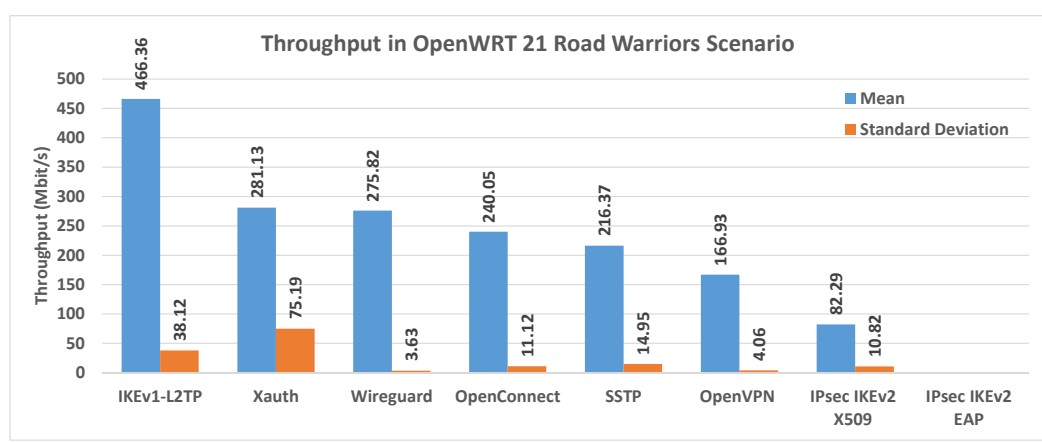

**Figure 19.** Comparison of Throughputs on OpenWRT 21 in all scenarios.

Figure 20 shows the comparison of the average throughput and standard deviation in a Road Warriors scenario, using a Windows 10 endpoint. The best results in terms of throughput are obtained here using IKEv2-EAP.

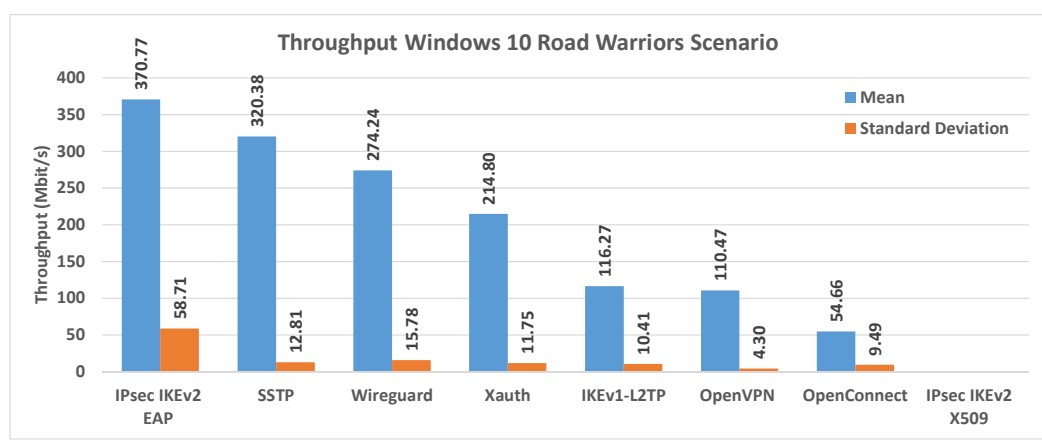

**Figure 20.** Comparison of Throughputs on Windows 10 in all scenarios.

*5.12. Experimental Evaluation of Different VPN Protocols on the Same Hardware*

Concerning the experimental evaluation of VPN protocols with different security levels on the same hardware, the choice was OpenVPN on the OpenWRT Router. OpenVPN in its most recent versions sets up two connections: a low-bandwidth "control channel" that negotiates network parameters and keys for the "data channel" and uses TLS to protect packets passing through it, and a "data channel", on which the actual VPN traffic travels, is encrypted with keys traded on the control channel. The version installed on OpenWRT uses TLS 1.3, the most recent, and the related data encryption algorithms were evaluated. In particular, the following testbeds were carried out, the results of which are highlighted in the following two graphs and summarized in Table 5.

**Table 5.** OpenVPN using different encryption algorithms configuring TLS-cipher suites and data-ciphers.

| TLS-CIPHER | DATA-CIPHER 1 | DATA-CIPHER 2 | DATA-CIPHER 3 |
|---|---|---|---|
| TLS-AES-128-GCM-SHA256 | AES-128-GCM | AES-256-GCM | CHACHA20-POLY1305 |
| TLS-AES-256-GCM-SHA384 | AES-128-GCM | AES-256-GCM | CHACHA20-POLY1305 |
| TLS-CHACHA20-POLY1305-SHA256 | AES-128-GCM | AES-256-GCM | CHACHA20-POLY1305 |

Figure 21 shows the throughput of the OpenVPN using different encryption algorithms. Figure 22 shows the RTT of the OpenWRT VPN using different encryption algorithms.

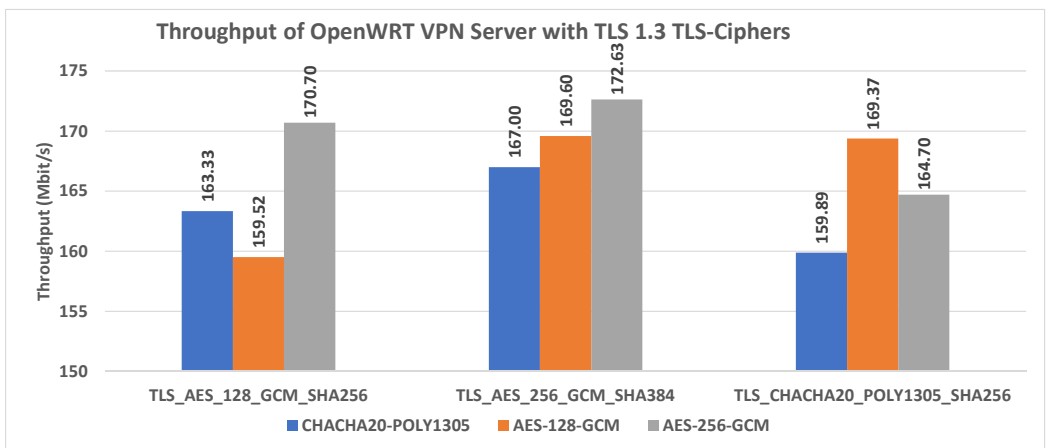

**Figure 21.** Throughput of OpenVPN using different encryption algorithms.

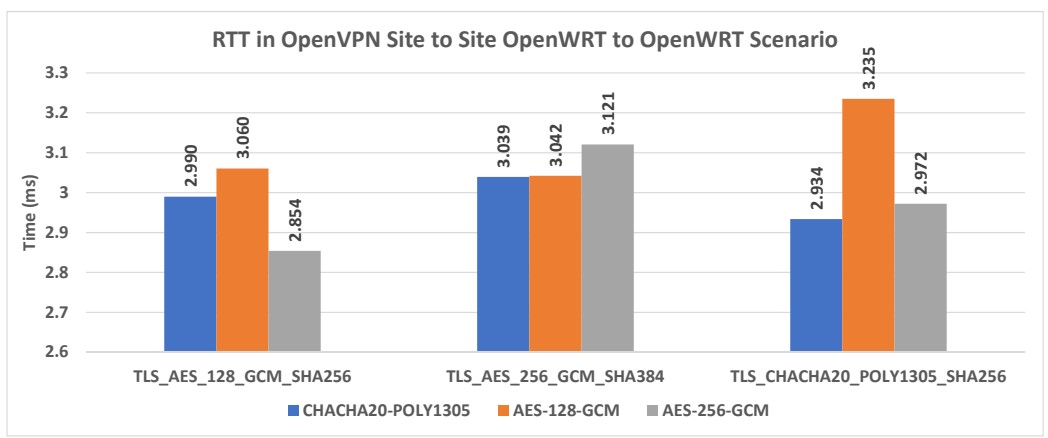

**Figure 22.** RTT of the OpenWRT VPN using different encryption algorithms.

In Figure 23 the PHY RATE data collected during the VPN connection tests with wireless devices are shown.

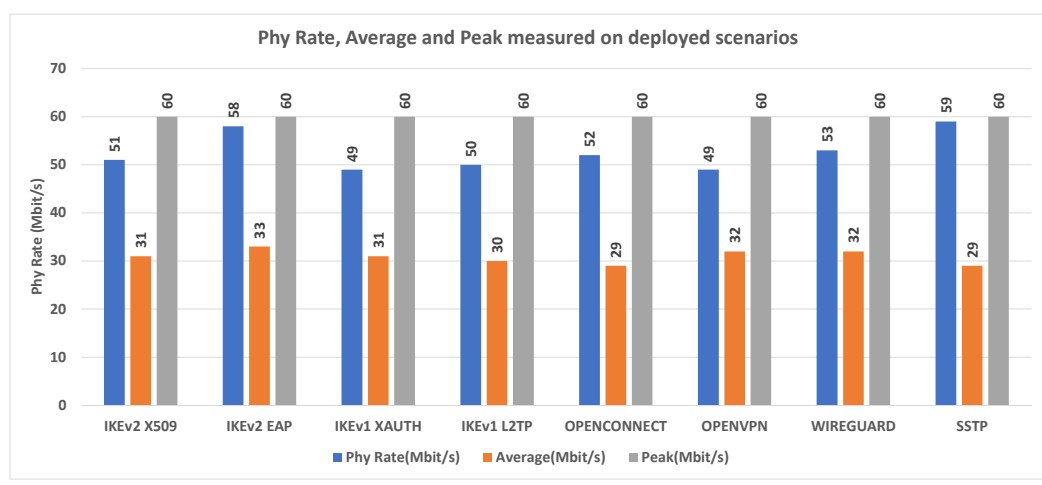

**Figure 23.** Phy Rate, Average and Peak measured on deployed scenarios.

In Figure 24 the RTT data collected during the VPN connection tests with wired devices are shown.

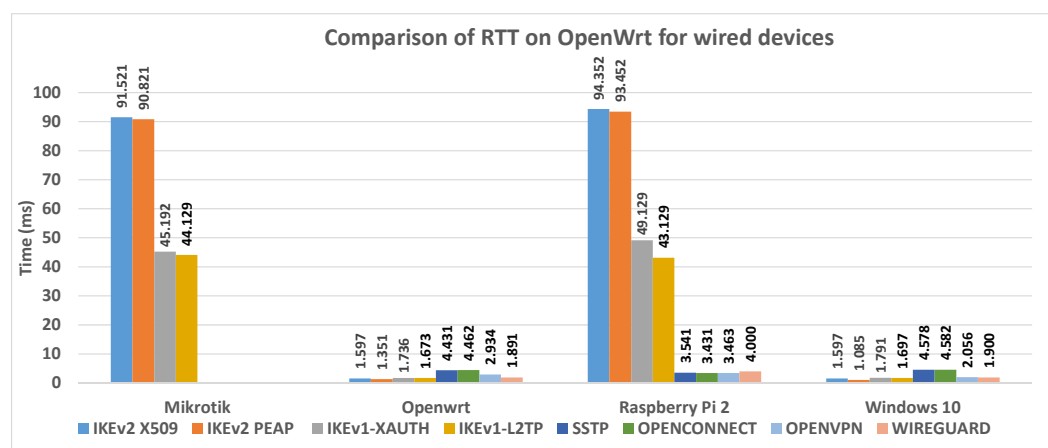

**Figure 24.** RTT data collected during the VPN connection tests with wired devices.

In Figure 25 the RTT data collected during the VPN connection tests with wireless devices are shown. Both test batteries expected 200 packets sent from one endpoint to another.

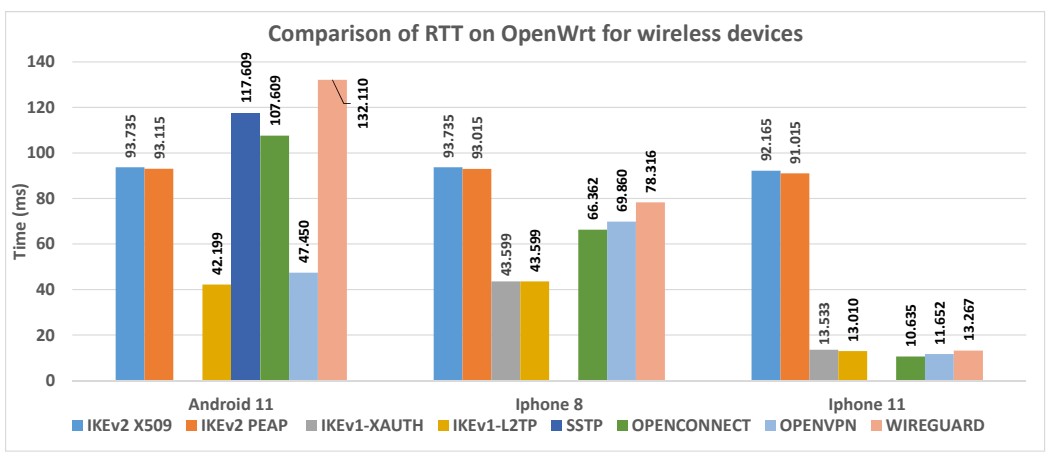

**Figure 25.** RTT data collected during the VPN connection tests with wireless devices.

From the comparative analysis of Figures 21 and 22, AES-256-GCM, using TLS-ciphers 1,3, turns out to be the best data cipher in terms of throughput and RTT, followed by AES-128-GCM and CHACHA20-POLY1035. From the analysis of Figure 23, OpenWRT results have an average Phy Rate ranging between 29 and 33 Mbps on the TP-LINK TL-WR841N device. From the comparative analysis of Figures 24–27, OpenWRT is the best device/OS for wired connections, while iPhone 11 is the best RoadWarrior wireless client.

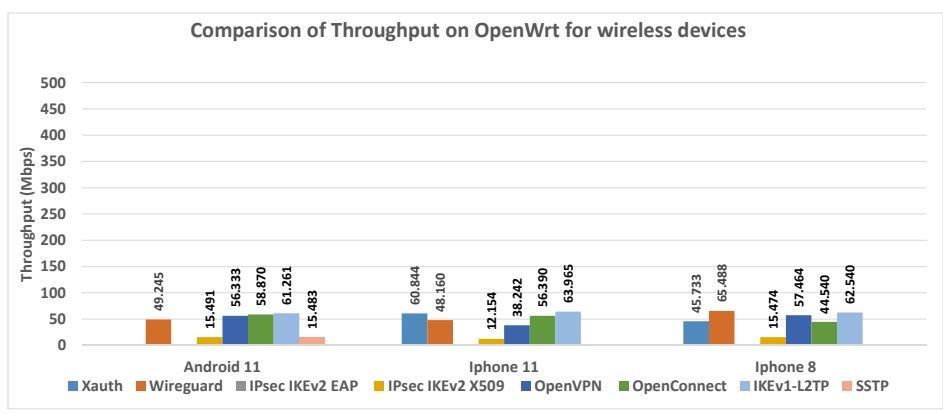

**Figure 26.** Throughputs data collected during the VPN connection tests with wireless devices.

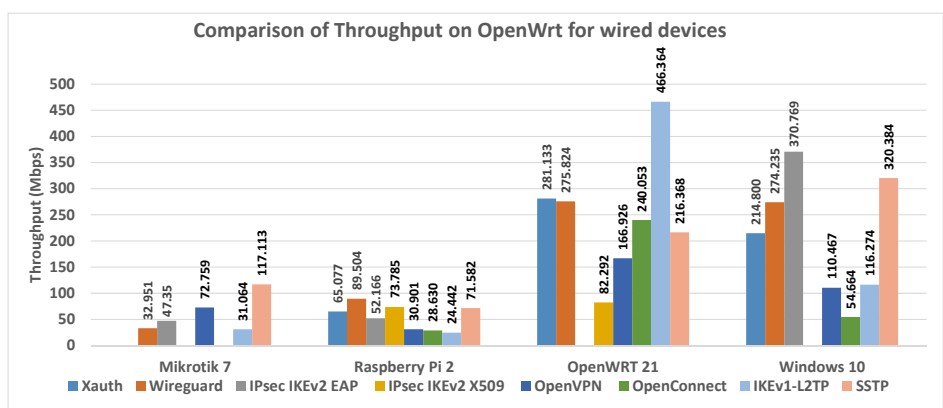

**Figure 27.** Throughputs data collected during the VPN connection tests with wired devices.

*5.13. Summary of Experimental Data Obtained for Deployed Scenarios*

In the deployed scenarios the practical goodness of choice based on OpenWRT was evaluated.

Making a grouping of the considered VPN deployed, we can make the following considerations: IKEv2-X509, IKEv2-PEAP and IKEv1-L2TP, having the best throughput performance, can represent the first implementation option. Then, the second group is represented by Wirecard, SSTP, and OpenConnect, and the third group is composed of OpenVPN and IKEv1-XAUTH. These considerations are made on the basis of the HW used for our experiments and shown in Table 3. Figure 23 illustrates Phy Rate, Average Phy Rate, and Phy Rate Peak measured on scenarios implemented for all wireless VPN clients, taking into account what is the reference Router (TP-LINK TL-WR841N), which uses 802.11n. Figure 24 illustrates RTT data collected during the VPN connection tests with wired and Figure 25 wireless devices. Tables 6 and 7 summarizes the result obtained for best data transmission efficiency algorithms of deployed scenarios and VPN algorithms which provide the best compatibility with the currently deployed infrastructures.

**Table 6.** Best data transmission efficiency algorithms of deployed scenarios.

| VPN TYPE | PHASE 1 CIPHERS | PHASE 2 CIPHERS |
|---|---|---|
| IKEv1-L2TP | aes256-sha1-sha256-sha384-modp2048 | aes128-aes256-sha1-sha256-modp2048-modp4096 |
| IKEv2-X509 | sha512-modp3072-aes256-modp2048s256 | sha512-modp3072-aes256-modp2048s256 |
| IKEv2-PEAP | sha512-modp3072-aes256-modp2048s256 | sha512-modp3072-aes256-modp2048s256 |

**Table 7.** VPN algorithms which provide the best compatibility with the current infrastructures.

| VPN TYPE | CLIENT NUMBERS | CIPHERS |
|---|---|---|
| IKEv1-L2TP | 7 DIFFERENT VENDORS | aes256-sha1-sha256-sha384-modp2048 aes128-aes256-sha1-sha256-modp2048-modp4096 |
| WIREGUARD | 8 DIFFERENT VENDORS | ChaCha20 / Curve25519 |
| OPENVPN | 8 DIFFERENT VENDORS | CHACHA20-POLY1305 / AES-128-GCM |

In Figure 28 one shows the comparison of throughput on OpenWrt in all scenarios. From Figure 28 it is possible to view the IKE VPN software has the best throughput values. Moreover, that IKEv1 which has lighter cryptography guarantees the best throughput. Also, the ranking shown in Figure 28 confirms that this firmware is a valid option for creating secure, robust networks, even with low-cost devices.

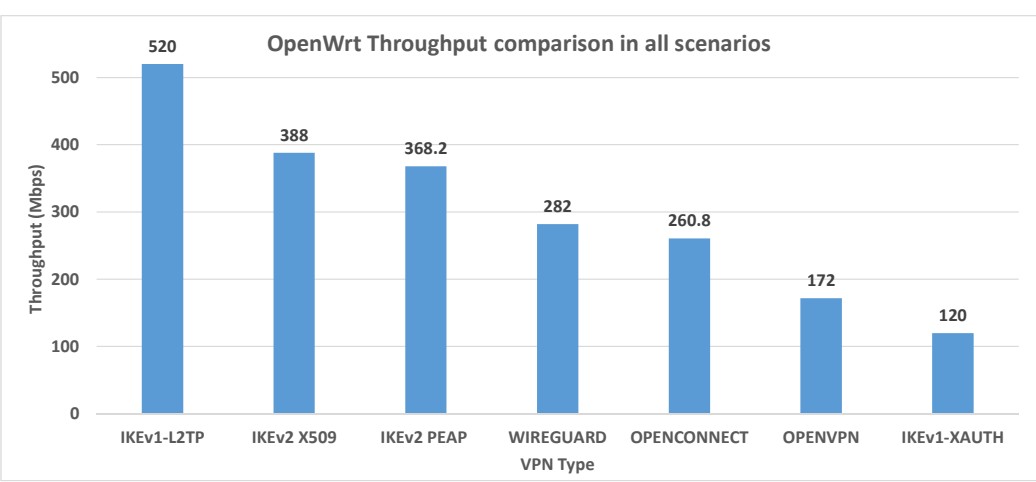

**Figure 28.** Comparison of Throughputs on OpenWrt in all scenarios.

## 6. Conclusions

To achieve the objectives of this work, OpenWRT open firmware has been chosen. This firmware has shown great potential, used in "enterprise" contests with custom HW, such as card APU ("APU BOARDS ", https://openwrt.org/toh/pcengines/apu2 (accessed on 10 June 2022)), and in "home" environments, with standard routers that support open firmware. These potentials are especially useful in outdoor environments, with limited connectivity and in areas difficult to wire (rural/mountain areas), and with the need to manage encrypted connections (often VPN) that cover more or less extensive sensor networks, even of a different nature. Using specific HW and installing dedicated SW, it is possible to directly interconnect these routers as active members of the network using different protocols such as local Zigbee networks interfaced through a Raspberry Pi gateway, like Lora/Lorawan networks, also linked to LTE/4G/5G networks. This firmware, loading a light version of the Linux system, allows having both a flexible operating system (the same router can be both multi VPN concentrator and MQTT broker, for example) and costs reduction by allowing the maximum exploitation of constrained HW. Various implemented scenarios demonstrate the quality of the throughput of these devices as well as their wide flexibility in the choice of the VPN protocols to be implemented. They can be both IPsec initiators/responders and members of TLS MESH over OpenVPN, OpenConnect, and Wireguard networks, achieving good performance in data exchange and resource use. In the scenarios created, by using Iperf, it was evaluated the practical goodness of choice based on OpenWRT, and from Figures 6–13 it turns out that both in client/initiator mode and server/responder mode, it continually ranks in the first three places for Transfer rate. This ranking confirms that this firmware is a valid option for creating secure networks, even with low-cost devices. Future implementations could foresee using more advanced platforms than APUs to increase this system's functionality. It may become a hypervisor, also, to create failover/load balancing policies in addition to that already present infrastructure to make the services perform better without upsetting the basic architecture. OpenWRT is the tool we use to answer all three goals of this paper as it supports all the VPN protocols and implementation deploys examined and it is itself the answer to reaching our third goal. As for the first goal of our work one aims to find the best algorithms that guarantee the best data transmission efficiency on constrained devices for each type of VPN deployed. The constrained devices considered are the TP-LINK OpenWRT router, the Mikrotik router, and the Raspberry Pi 2. The best data transmission efficiency algorithms, from the experiments carried out, appear to be those related to: IKEv1-L2TP, IKEv2-X509, and IKEv2-PEAP. As for the second goal of our work, VPN algorithms that provide best compatibility with the current infrastructures, from the experiments carried out, appear to be those related to: OpenVPN, Wireguard, and IKEV1-L2TP. For that concern VPN, deployed with different security levels on the same hardware, the experimentation proves that OpenVPN, using

TLS 1.3 with different data encryption algorithms, on OpenWRT router results in the best choice, as shown in the discussion of experimentation results. Further analyses with all shown VPNs could be the subject of future works.

**Author Contributions:** Conceptualization, A.F.G., F.D.R. and D.M.; methodology, M.T.; software, A.F.G.; validation, A.F.G., D.M. and E.G.; data curation, A.F.G. and D.M.; writing—original draft preparation, A.F.G., M.T. and E.G.; writing—review and editing, A.F.G. and M.T.; supervision, F.D.R. All authors have read and agreed to the published version of the manuscript.

**Funding:** This research received no external funding.

**Data Availability Statement:** Not applicable.

**Conflicts of Interest:** The authors declare no conflict of interest.

## Abbreviations

The following abbreviations are used in this manuscript:

| | |
|---|---|
| AES | Advanced Encryption Standard |
| AH | Authentication Header |
| EAP | Extensible Authentication Protocol |
| ESP | Encapsulating Security Payload |
| GDPR | General data protection regulation |
| HW | Hardware |
| IETF | Internet Engineering Task Force |
| IKE | Internet Key Exchange |
| IoT | Internet of Things |
| IPsec | IP Security |
| ISAKMP | Internet Security Association and Key Management Protocol |
| MQTT | Message Queue Telemetry Transport |
| NAT | Network Address Translation |
| OTP | On-Time Password |
| PSK | Pre Shared Key |
| PSTN | Public Switched Telephone Network |
| RTT | Round Trip Time |
| RW | Road Warriors |
| SAD | Security Association Database |
| SPD | Security Policy Database |
| SSL | Secure Sockets Layer |
| SW | Software |
| TLS | Transport Layer Security |
| UTP | Unshielded Twisted Pair |
| VPN | Virtual Private Network |

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
