# Peer review of "A VPN Performances Analysis of Constrained Hardware Open Source Infrastructure Deploy in IoT Environment"

_futureinternet, doi:10.3390/fi14090264_

Round 1
Reviewer 1 Report
The paper deals with interesting and current topics in more practical than scientific manner. But the form of the paper is poor. Chaotic organization, many improper words, lack of clear figures, and wrong design of experiments.
The paper must be corrected and new experiments should be performed.
Chaotic order of sections:
» sections 1 and 2 should be reorganized: Introduction, Description of different VPN solutions, Description of experiment
» section 3 (Related works) has the subsection 3.1 (Main contribution of the paper), which is not "related work"
» section 4 shows Experimental results, but the experiment has not been properly described
Important issues:
» Table 4 contains a list of devices used to create test environment. Some of the tested devices (at least the iPhones) are probably wireless-only devices. Looking at presented charts, I can pretend, that these devices were connected to quite old TP-Link TL-WR841N which only operates in IEEE 802.11n on 2.4 GHz band. The comparison of their results with results of devices connected by wire is useless.
» Why the relatively old equipment has been chosen? Raspberry Pi 2, TL-WR841N and fast-ethernet switch? I think that the usage of IEEE 802.11n on 2.4GHz band and 100 Mbps Ethernet has affected the results.
» The description of architecture starts on line 325. I know already that it is not complete. And it was hard for me to imagine how the devices were connected. What is the subnet representing WAN? How the two private subnets were connected? Please draw some schematic view of the experiments.
» Why iperf has been used in two versions? Did the results change? The results presented later on the charts were measured with iperf2 or iperf3?
» It is also a good idea to present more results from the experiments. The throughput over time charts does not tell much about the results. I think the interesting parameters are: mean throughput, its STD, RTT. For the wireless connected devices authors should also write the wireless connection parameters (PHY layer bitrate) at least.
» How the comparison in table 5 was made? What is "high" security? "Low" cost etc? This is only author's opinion and there is no text in the paper to describe how it has been graded.
Presentation and language:
» Lines 345-349 are doubled
» Legends and axes are barely readable on all the charts
» Throughput on the Y-axes is presented in megabytes per second instead of megabits per second
» Bit Rates and Throughput are not the same thing. Please correct the title of figures.
» Wireguard is not the VPN service (434). Service is not protocol. Protocol is not service. Also protocol in not the network. You can't write "SSTP is VPN". You should write "SSTP is VPN protocol". Please do not oversimplify the text.
Reviewer 2 Report
Paper “A VPN Performances Analysis of constrained Hardware Open Source infrastructure deploy in IoT Environment” presents results of experimental evaluation of various VPN technologies trying to answer to three main questions: what is the best algorithm which guarantees best data transmission efficiency on constrained devices; what are the VPN algorithms which provide best compatibility with the current infrastructures; and what are the best capabilities of open firmware usage on constrained devices (routers and firewalls).
The paper still requires some additional corrections before it could be published.
-
Some references are very general (lines 514-516, 519-521, 557). I think more specific and more “scientific” review paper should be used.
-
The quality of Figures 1-3 should be improved. Now it is not possible to read the text.
-
The Table 1 should be clarified. Does “SITE TO SITE” is the same application scenario as “SSTP SITE TO SITE” or “IKEv1 SITE TO SITE PSK”? This should be clarified separating the names of the scenarios used by their implementers from the application areas of each implementation.
-
I do not agree that SSL/TLS is the VPN protocol. Sure, it could be used (and is used) to implement VPNs, but was not created as VPN protocol per se (line 90).
-
The meaning of Table 2 is somehow not clear. Do these settings of the protocols represent settings used in the experiments? If so, these should be placed in appropriate chapter. If not, then the meaning of the table should be explained in more detail.
-
Conclusions should clearly present answers to all three questions introduced in the beginning of the paper (lines 11-17). Now only the conclusion about OpenWRT firmware are presented in very high detail.
-
Some experiments are not correct, as authors are comparing different versions of the protocols on different hardware. In Figure 11 AES-256-GCM-SHA384 cipher suite is compared with AES-128-CBC. In Figure 7 AES-CBC-128 is compared with AES-CBC-256. Although the experimental evaluation of VPN protocols with different security levels on the SAME hardware would benefit the value of the paper. I. e. it would be very interesting to compare the performance of e. g. OpenVPN using different security methods for confidentiality and integrity on the same router hardware.
Round 2
Reviewer 1 Report
» The sentence starting on line 38 is confusing. I suppose it should describe the encapsulation in VPN. Why then only the first packet is encapsulated? Is it the requirement that only non-routable (what does this mean?) and private network packets are encapsulated?
» Please describe what is the "dummy WAN" (line 375).
» Figures 6-13 - Why the STD values are presented as a function of the device type (connected points)? It does not make any sense. Do not connect them. It would also be better if the contrast between the elements of the bars on the figures is higher.
» It is still difficult for me to understand, why completely different devices are compared. I mentioned this in my previous review. Probably I have not stated it enough. I understand the authors wanted to show which VPN protocol is the most effective. But comparing it with different devices does not tell us anything. We have to compare different VPNs on the same machines, not different machines running the same protocol. And we need to know what is the throughput without using any VPN protocol. The authors have the data, but the data should be presented not by grouping them by the protocols.
» I have found a few misspellings of Raspberry Pi name.
» Please use spell check before submitting the next version. I also suggest using automated tools, like Grammarly, to rephrase some of the sentences.
Reviewer 2 Report
I think the manuscript is now good for publishing as the authors have significantly improved it.
Round 3
Reviewer 1 Report
The paper, in its recent version, can be published. All the problems in the paper have been fixed.